# Convex Optimization with Local Label Differential Privacy: Tight Bounds in All Privacy Regimes

**Lynn Chua**                                                      *chualynn@google.com*
*Google Research*
**Badih Ghazi**                                                    *badihghazi@gmail.com*
*Google Research*
**Ravi Kumar**                                                     *ravi.k53@gmail.com*
*Google Research*
**Pasin Manurangsi**                                              *pasin@google.com*
*Google Research*
**Ziteng Sun**                                                     *zitengsun@google.com*
*Google Research*
**Chiyuan Zhang**                                                  *chiyuan@google.com*
*Google Research*

**Reviewed on OpenReview:** *https: // openreview. net/ forum? id= 8SjmOFrV2u*

## Abstract

We study the problem of Stochastic Convex Optimization (SCO) under the constraint of local Label Differential Privacy (L-LDP). In this setting, the features are considered public, but the corresponding labels are sensitive and must be randomized by each user locally before being sent to an untrusted analyzer. Prior work for SCO under L-LDP (Ghazi et al., 2021) established an excess population risk bound with a *linear* dependence on the size of the label space, $K$: $O\left(\frac{K}{\varepsilon\sqrt{n}}\right)$ in the high-privacy regime ($\varepsilon \leq 1$) and $O\left(\frac{K}{e^\varepsilon\sqrt{n}}\right)$ in the medium-privacy regime ($1 \leq \varepsilon \leq \ln K$). This left open whether this linear cost is fundamental to the L-LDP model. In this note, we resolve this question. First, we present a novel and efficient non-interactive L-LDP algorithm that achieves an excess risk of $O\left(\sqrt{\frac{K}{\varepsilon n}}\right)$ in the high-privacy regime ($\varepsilon \leq 1$) and $O\left(\sqrt{\frac{K}{e^\varepsilon n}}\right)$ in the medium-privacy regime ($1 \leq \varepsilon \leq \ln K$). This quadratically improves the dependency on the label space size from $O(K)$ to $O(\sqrt{K})$. Second, we prove a matching information-theoretic lower bound across all privacy regimes for any sufficiently large $n$.

## 1 Introduction

The proliferation of large-scale data collection and analysis has created a fundamental tension between extracting statistical insights for machine learning and safeguarding the privacy of individuals contributing the data. Differential Privacy (DP) (Dwork et al., 2006; Dwork & Roth, 2014) has emerged as a gold standard for private data analysis, as it provides a mathematically rigorous and provable guarantees: a randomized algorithm satisfies $\varepsilon$-DP if its output distribution is nearly invariant to the addition or removal of any single individual's data point. This provably bounds the privacy loss for any individual, regardless of the adversary's computational power or auxiliary knowledge.

While offering desired protection, the stringent guarantees of standard DP can impose a significant cost on statistical utility, often leading to a challenging privacy-utility tradeoff. In many practical machine learning applications, this guarantee may be unnecessarily strong—it is common for the features $X_i$ of a dataset to be public or non-sensitive, while the corresponding labels $Y_i$ contain the sensitive, private information. For instance, in medical machine learning, a patient's demographic information (features) might be semi-public,

but their diagnosis or medical conditions (the label) is highly confidential (Tang et al., 2022). This common scenario motivates the study of *Label DP* (Chaudhuri & Hsu, 2011), a well-established relaxation where privacy is guaranteed only with respect to changes in the labels of the training examples[1]:

**Definition 1** (Label DP). An algorithm $\mathcal{A}$ is said to be $\varepsilon$-*Label DP* if for all input datasets $D = \{(X_i, Y_i)\}_{i \in [n]}$ and $D' = \{(X_i', Y_i')\}_{i \in [n]}$ that differ by at most one label of a single data point and for all possible output $O$, we have $\Pr(\mathcal{A}(D) = O) \leq e^\varepsilon \cdot \Pr(\mathcal{A}(D') = O)$.

In this work, we study Label DP in the context of Stochastic Convex Optimization (SCO). Given $n$ i.i.d. samples $\{(X_i, Y_i)\}_{i=1}^n$ from an unknown distribution $P$ over $\mathcal{Z} := \mathcal{X} \times [K]$, the goal is to find a parameter $\hat{w}$ that minimizes the *population loss* $\mathcal{L}(w, P) := \mathbb{E}_{(X,Y) \sim P}[\ell(w, (X, Y))]$ (Shalev-Shwartz et al., 2009). Our objective is to bound the excess population risk, $\mathbb{E}[\mathcal{L}(\hat{w}, P)] - \mathcal{L}(w^*, P)$, where $w^* = \arg\min_{w \in \mathcal{W}} \mathcal{L}(w, P)$. We assume throughout that the loss function $\ell$ is convex and $L$-Lipschitz (w.r.t. $w$), and the parameter $w$ comes from parameter space $\mathcal{W}$ with diameter at most $R$.

We consider SCO under the stringent *local* model (Kasiviswanathan et al., 2011). Unlike the central DP model, which assumes a trusted data curator, local DP provides stronger privacy by requiring each user to randomize their own data locally before transmitting it to an untrusted central analyzer. This combination defines our problem: SCO under *local Label DP* (L-LDP), where each user $i$ possesses a public feature $X_i$ and a private label $Y_i$, and sends a sanitized version of their label $S_i = \mathcal{R}_i(Y_i)$ to the analyzer, where $\mathcal{R}_i$ is an $\varepsilon$-L-LDP randomizer (see Definition 2)

SCO with Label DP has been studied in several recent papers (Ghazi et al., 2021; 2024a;b). For the local DP model, Ghazi et al. (2021) provide an algorithm based on Randomized Response that satisfies $\varepsilon$-L-LDP and achieves an excess risk bound[2] of $O\left(\max\left\{\frac{K}{e^\varepsilon - 1}, 1\right\} \cdot \frac{RL}{\sqrt{n}}\right)$. More concretely, the excess risk bounds in the three regimes of parameters are:

- $O\left(\frac{K}{\varepsilon} \cdot \frac{RL}{\sqrt{n}}\right)$ for the *high*-privacy regime where $\varepsilon \leq 1$,
- $O\left(\frac{K}{e^\varepsilon} \cdot \frac{RL}{\sqrt{n}}\right)$ for the *medium*-privacy regime where $1 \leq \varepsilon \leq \ln K$,
- $O\left(\frac{RL}{\sqrt{n}}\right)$ for the *low*-privacy regime where $\varepsilon \geq \ln K$.

Note that in low-privacy regime, the excess risk is the same as for non-private algorithms, i.e., there is no (asymptotic) cost of Label DP. In the other two regimes, however, the excess risk depends *linearly* on the number of labels $K$, which can present a significant barrier to scalability for problems with large label spaces. Note that Ghazi et al. (2021) provide another algorithm based on Gaussian noise but with $\sqrt{K}$ dependency; however, it only satisfies[3] $(\varepsilon, \delta)$-L-LDP (aka *approximate*-DP) rather than $\varepsilon$-L-LDP (*pure*-DP) discussed so far. Still, this bound is notably worse than what can be achieved in the trusted *central* Label DP model, where the dependency on $K$ is only polylogarithmic (Ghazi et al., 2024b). This leaves a critical open question: Is this polynomial dependency on the label space size an unavoidable, fundamental cost of the local privacy model, or is it an artifact of existing techniques?

## 1.1 Our Contributions

In this note, we resolve this question by providing new, tight bounds for SCO under L-LDP.

---

[1]It should be stressed that Label DP does *not* provide any privacy protection for the features. As such, Label DP only limits an adversary's ability to infer the label beyond what is possible for an adversary with access to the features. For example, if each label is completely determined by the features, then the adversary will be able to exactly identify the label.

[2]Strictly speaking, Ghazi et al. (2021) only state their bounds for $\varepsilon \leq O(1)$ case. However, it is easy to generalize this to the other two cases via calculations similar to Section A with $d = 1$.

[3]During the preparation of this manuscript, we observe that, instead of using Gaussian noise, we may use the vector summation mechanism from (Duchi et al., 2013) to achieve $O\left(\frac{\sqrt{K}}{\varepsilon} \cdot \frac{RL}{\sqrt{n}}\right)$ excess risk with $\varepsilon$-L-LDP in the low-privacy regime. However, the risk remains $O\left(\sqrt{K} \cdot \frac{RL}{\sqrt{n}}\right)$ in the medium-privacy regime, unlike our algorithm in Theorem 1. See Section B for a more detailed discussion.

Our first contribution is an efficient L-LDP algorithm (Algorithm 1), based on a subset selection randomization mechanism (Ye & Barg, 2018; Wang et al., 2016), which achieves an improved excess risk bound:

**Theorem 1.** *There exists a non-interactive local $\varepsilon$-label DP algorithm $\hat{w}$ satisfying the following*

$$\mathbb{E}_{\hat{w} \leftarrow \mathcal{A}(P^{\otimes n})}\left[\mathcal{L}(\hat{w}, P)\right] - \mathcal{L}(w^*, P) = O\left(\sqrt{\max\left\{\frac{Ke^\varepsilon}{(e^\varepsilon - 1)^2}, 1\right\}} \cdot \frac{RL}{\sqrt{n}}\right)$$

$$= \frac{RL}{\sqrt{n}} \cdot \begin{cases} O\left(\frac{\sqrt{K}}{\varepsilon}\right) & \text{if } \varepsilon \leq 1, \\ O\left(\sqrt{\frac{K}{e^\varepsilon}}\right) & \text{if } 1 \leq \varepsilon \leq \ln K, \\ O(1) & \text{if } \varepsilon \geq \ln K. \end{cases}$$

In the low- and medium-privacy regimes, our bound quadratically improves the dependency on the label space size from $O(K)$ in prior work (Ghazi et al., 2021) to $O(\sqrt{K})$. Our algorithm is non-interactive, meaning that each user can provide their sanitized label immediately without needing to receive any additional information.

Our second contribution is to prove that this bound is, in fact, optimal. We establish a new information-theoretic lower bound for any algorithm satisfying $\varepsilon$-L-LDP for the SCO problem, as stated below. In fact, our lower bound holds even for the more relaxed fully-interactive setting. (See Section 2 for formal definitions of non-interactive and fully-interactive algorithms.)

**Theorem 2.** *Let $\varepsilon > 0$ and $K, n \in \mathbb{N}$ be such that $n \geq \Theta\left(\frac{e^\varepsilon K^2}{(e^\varepsilon - 1)^2}\right)$. Then, for any fully-interactive $\varepsilon$-local DP algorithm $\mathcal{A}$, there exists a distribution $P$ on $\mathcal{X} \times [K]$ such that,*

$$\mathbb{E}_{\hat{w} \leftarrow \mathcal{A}(P^{\otimes n})}\left[\mathcal{L}(\hat{w}, P)\right] - \mathcal{L}(w^*, P) = \Omega\left(\sqrt{\max\left\{\frac{Ke^\varepsilon}{(e^\varepsilon - 1)^2}, 1\right\}} \cdot \frac{RL}{\sqrt{n}}\right).$$

Our upper and lower bounds match up to constant factors across all privacy regimes ($\varepsilon$) for any sufficiently large $n$. Thus, our results completely characterize the minimax optimal excess risk for SCO under L-LDP. This demonstrates that the true worst-case cost of L-LDP for this problem scales as $\Theta(\sqrt{K})$ in the low- and medium-privacy regimes.

## 2 Preliminaries

We recall formal definitions of local DP randomizers and non-interactive/fully-interactive local label DP below. These definitions follow their standard DP counterpart (e.g., (Duchi et al., 2018; Joseph et al., 2019)). Note that, in all definitions below, we implicitly assume that $X_1, \ldots, X_n$ are publicly known and thus the randomizers can depend arbitrarily on them.

**Definition 2** (Local Label DP (L-LDP) Randomizer). *A randomized algorithm $\mathcal{R} : [K] \to \mathcal{S}$ is $\varepsilon$-L-LDP if for any labels $y, y' \in [K]$ and any $S \in \mathcal{S}$, $\Pr(\mathcal{R}(y) = S) \leq e^\varepsilon \cdot \Pr(\mathcal{R}(y') = S)$.*

The non-interactive setting is where each user has to apply their local randomizer without seeing the output of any other randomizer. This is the most stringent setting of local DP. Our algorithm works in this model.

**Definition 3** (Non-Interactive L-LDP). *An algorithm $\mathcal{A}$ satisfies non-interactive $\varepsilon$-L-LDP if it consists of a set $\{\mathcal{R}_i\}_{i=1}^n$ of $n$ local randomizers and an analyzer $\mathcal{A}_0$ such that for each data point $i \in [n]$ with private label $Y_i$:*

1. *The local randomizer $\mathcal{R}_i : [K] \to \mathcal{S}$ is an $\varepsilon$-L-LDP randomizer.*
2. *A sanitized label $S_i = \mathcal{R}_i(Y_i)$ is generated locally by the $i$th user, independent of all other data points and outputs.*

The final output is computed by $\mathcal{A}_0$ using only the public features and the sanitized labels:

$$\mathcal{A}(\{(X_i, Y_i)\}_{i \in [n]}) = \mathcal{A}_0(\{(X_i, S_i)\}_{i \in [n]}).$$

The fully-interactive setting is the most relaxed setting of local Label DP, where each user can now communicate with the analyzer in an arbitrary manner subject only to the constraint that the transcript satisfies Label DP.

**Definition 4** (Fully Interactive L-LDP)**.** An interactive protocol $\mathcal{A}$ between an aggregator and $n$ users is *fully interactive* $\varepsilon$-L-LDP if for any two datasets $D = \{(X_i, Y_i)\}_{i \in [n]}$ and $D' = \{(X_i, Y_i')\}_{i \in [n]}$ that differ by a single label, and for any possible transcript $T$ (where a transcript consists of all aggregator queries and user responses), we have

$$\Pr[\text{Transcript}(\mathcal{A}, D) = T] \le e^{\epsilon} \cdot \Pr[\text{Transcript}(\mathcal{A}, D') = T].$$

In this setting, the interactions can be arbitrary; the aggregator may adaptively choose which user to query next (potentially querying the same user multiple times) and what query to send based on the entire history of the interaction.

## 3    Label DP Algorithm for SCO

Our algorithm is based on (projected) SGD where we randomize each label at the beginning and use an unbiased gradient estimator in each iteration; so far, these are the same ingredients as in (Ghazi et al., 2021). The main difference is that, instead of using $K$-ary randomized response, we use a randomizer that is inspired by sample-optimal randomization algorithms under local DP for histogram estimation (Erlingsson et al., 2014; Wang et al., 2016; Ye & Barg, 2018; Feldman & Talwar, 2021). However, we modify[4] these randomizers (and estimators) slightly for ease of variance computation, as specified below.

*Proof of Theorem 1.* We start by defining the $\varepsilon$-L-LDP randomizer $\mathcal{R}$ that maps the label space $[K]$ to the output space $\mathcal{S}$, where $\mathcal{S} = 2^{[K]}$ is the power set of the label space. For each data point $t \in [n]$ with private label $Y_t$, the *local randomizer* $\mathcal{R}_t : [K] \to \mathcal{S}$ generates a *sanitized* label $S_t = \mathcal{R}_t(Y_t)$ independently. More specifically, each $i \in [K]$ is included in $S_t$ independently with probability

$$\Pr\left(i \in S_t\right) = \begin{cases} \frac{1}{2}, & \text{if } i = Y_t, \\ \frac{1}{e^{\varepsilon}+1}, & \text{if } i \ne Y_t. \end{cases} \tag{1}$$

It can be verified that for any $y \ne y'$ and set $S \in \mathcal{S}$, we have

$$\frac{\Pr\left(\mathcal{R}(y) = S\right)}{\Pr\left(\mathcal{R}(y') = S\right)} = \prod_{i \in S} \frac{\Pr\left(i \in \mathcal{R}(y)\right)}{\Pr\left(i \in \mathcal{R}(y')\right)} \cdot \prod_{i \notin S} \frac{\Pr\left(i \notin \mathcal{R}(y)\right)}{\Pr\left(i \notin \mathcal{R}(y')\right)} = \begin{cases} 1 & \text{if } y, y' \in S \text{ or } y, y' \notin S, \\ e^{\varepsilon} & \text{if } y \in S, y' \notin S, \\ e^{-\varepsilon} & \text{if } y \notin S, y' \in S. \end{cases}$$

Thus, the randomizer is $\varepsilon$-L-LDP as desired.

Next we describe the analyzer $\mathcal{A}_0$ for SCO under label DP.

Note that the overall algorithm $\mathcal{A}$ satisfies $\varepsilon$-label DP because it consists of $\varepsilon$-L-LDP local randomizers $\mathcal{R}_t$ generating sanitized labels $S_t$. We next prove that $\hat{g}_t$ is an unbiased estimate of $\nabla \ell(w_{t-1}, (X_t, Y_t))$, and hence is also an unbiased estimate of $\nabla \mathcal{L}(w_{t-1}, P)$ since $(X_t, Y_t) \sim P$. This holds since

$$\mathbb{E}[\hat{g}_t] = \frac{2(e^{\varepsilon}+1)}{e^{\varepsilon}-1} \left( \sum_{k \in [K]} \left( \Pr\left(k \in S_t\right) - \frac{1}{e^{\varepsilon}+1} \right) \cdot \nabla \ell(w_{t-1}, (X_t, k)) \right)$$

$$= \frac{2(e^{\varepsilon}+1)}{e^{\varepsilon}-1} \left( \sum_{k \in [K]} \left( \frac{1}{e^{\varepsilon}+1} \mathbb{1}\left\{k \ne Y_t\right\} + \frac{\mathbb{1}\left\{k = Y_t\right\}}{2} - \frac{1}{e^{\varepsilon}+1} \right) \cdot \nabla \ell(w_{t-1}, (X_t, k)) \right)$$

$$= \nabla \ell(w_{t-1}, (X_t, Y_t)).$$

---

[4]It is in fact also possible to use the randomizers from (Wang et al., 2016; Ye & Barg, 2018) directly (and achieve asymptotically the same excess risk) but the computation is more complicated. We sketch the argument in Section A.

---

**Algorithm 1** Label DP SCO with Privatized Label Subset.

---

**Input:** Dataset $\{(X_1, Y_1), \ldots, (X_n, Y_n)\}$
**Parameters:** Label DP parameter $\varepsilon$; diameter $R$; Lipschitz parameter $L$
 1: {Local randomization stage for each user $t$}
 2: **for** $t = 1, \ldots, n$ **do**
 3:   $S_t \leftarrow \mathcal{R}_t(Y_t)$                                          {Locally compute using (1)}
 4:   Send $(X_t, S_t)$ to $\mathcal{A}_0$                         {Send sanitized label to analyzer}
 5: {Learning stage}
 6: Select an arbitrary initial value $w_0 \in \mathcal{W}$
 7: $\eta \leftarrow \frac{R}{L} \sqrt{\frac{(e^\varepsilon - 1)^2}{2n(K + e^\varepsilon)e^\varepsilon}}$                                     {Learning rate}
 8: **for** $t = 1, \ldots, n$ **do**
 9:   For $k \in [K]$, compute $\nabla \ell(w_{t-1}, (X_t, k))$
10:   $\hat{g}_t \leftarrow \dfrac{2(e^\varepsilon + 1)}{e^\varepsilon - 1} \left( \displaystyle\sum_{k \in [K]} \left( \mathbb{1}\{k \in S_t\} - \frac{1}{e^\varepsilon + 1} \right) \cdot \nabla \ell(w_{t-1}, (X_t, k)) \right)$
11:   $w_t \leftarrow \text{Proj}_{\mathcal{W}}(w_{t-1} - \eta \hat{g}_t)$                         {Project and update}
12: **return** $\bar{w} \leftarrow \frac{1}{n} \sum_{t=1}^{n} w_t$

---

Moreover, since the events $\mathbb{1}\{k \in S_t\}$ are independent, we have

$$
\begin{aligned}
\mathbb{E}\left[\|\hat{g}_t\|^2\right] =& \mathbb{E}\left[\|\nabla \ell(w_{t-1}, (X_t, Y_t))\|^2\right] + \mathbb{E}[\|\hat{g}_t - \nabla \ell(w_{t-1}, (X_t, Y_t))\|^2] \\
\leq& L^2 + \left(\frac{2(e^\varepsilon + 1)}{e^\varepsilon - 1}\right)^2 \sum_{k \neq Y_t} \left( \|\nabla \ell(w_{t-1}, (X_t, k))\|^2 \cdot \mathbb{E}\left[ \left( \mathbb{1}\{k \notin S_t\} - \frac{1}{e^\varepsilon + 1} \right)^2 \right] \right) \\
&+ \left(\frac{2(e^\varepsilon + 1)}{e^\varepsilon - 1}\right)^2 \|\nabla \ell(w_{t-1}, (X_t, Y_t))\|^2 \cdot \mathbb{E}\left[ \left( \mathbb{1}\{Y_t \in S_t\} - \frac{1}{2} \right)^2 \right] \\
=& L^2 \left( 1 + \left(\frac{2(e^\varepsilon + 1)}{e^\varepsilon - 1}\right)^2 \left( (K - 1) \cdot \frac{e^\varepsilon}{(e^\varepsilon + 1)^2} + \frac{1}{4} \right) \right) \\
\leq& L^2 \left( 1 + \frac{4e^\varepsilon(K + e^\varepsilon)}{(e^\varepsilon - 1)^2} \right).
\end{aligned}
$$

Hence by standard analysis of projected stochastic gradient descent for convex functions (e.g., (Bubeck, 2015, Section 6.1)), we have

$$
\mathcal{L}(\bar{w}, P) - \mathcal{L}(\bar{w}, w^*) \leq \frac{R^2}{\eta n} + \frac{\eta}{2n} \sum_{t=1}^{n} \mathbb{E}\left[\|\hat{g}_t\|^2\right] \leq \frac{R}{\eta n} + \frac{\eta L^2}{2} \left( 1 + \frac{4e^\varepsilon(K + e^\varepsilon)}{(e^\varepsilon - 1)^2} \right).
$$

Plugging in the choice of $\eta$ from Algorithm 1, we get the desired bound. $\qquad\square$

## 4 Lower Bound

For $\theta \in [0, 1]^K$ such that $\sum_{i \in [K]} \theta_i = 1$, we let $\mathcal{D}_\theta$ be the distribution on $[K]$ with probability mass $\theta_i$ at each $i \in [K]$. For simplicity, for a set $A$, we use $\theta \in A$ to denote that for each $i \in [K], \theta_i \in A$. We will reduce from the discrete distribution estimation problem with $\ell_2$-error. In the *discrete distribution estimation* problem, there is an unknown underlying distribution $\mathcal{D}_\theta$ on $[K]$. Each user $i$ receives $Y_i \in [K]$ and the goal is to estimate $\theta$, where the utility is measured in terms of the expected $\ell_2$-error. We use the following lower bound from (Acharya et al., 2023, Corollary 4)[5].

---

[5]While the main body of (Acharya et al., 2023) only states this lower bound for the more restricted *sequentially-interactive* local DP, their Appendix A shows how to extend this to the fully-interactive setting.

**Theorem 3** ((Acharya et al., 2023))**.** *Let $\varepsilon > 0$ and $K, n \in \mathbb{N}$ be such that $n \geq \Theta\left(\frac{e^{\varepsilon}K^2}{(e^{\varepsilon}-1)^2}\right)$. Then, for any fully-interactive $\varepsilon$-local DP algorithm $\mathcal{A}'$ for discrete distribution estimation, there exists a distribution $\mathcal{D}_{\theta}$ such that*

$$\mathbb{E}_{\hat{\theta} \leftarrow \mathcal{A}'(\mathcal{D}_{\theta}^{\otimes n})}[\|\hat{\theta} - \theta\|_2] \geq \Omega\left(\sqrt{\frac{e^{\varepsilon}K}{(e^{\varepsilon}-1)^2}} \cdot \frac{1}{\sqrt{n}}\right).$$

*Moreover, if $K$ is even, this holds even when $\theta \in \left\{\frac{1}{K} + \gamma, \frac{1}{K} - \gamma\right\}$ for some $\gamma \in \left[0, \frac{1}{2K}\right]$ such that $\gamma = \Theta\left(\sqrt{\frac{e^{\varepsilon}}{(e^{\varepsilon}-1)^2}} \cdot \frac{1}{\sqrt{n}}\right)$ and $\gamma$ is known to the algorithm $\mathcal{A}'$.*

Our proof of Theorem 2 proceeds by setting the loss as a linear loss that ignores the feature vector and attempts to learn the parameter $\theta$ of the label distribution $\mathcal{D}_{\theta}$. We remark that similar ideas have been used in the literature (e.g., (Bassily et al., 2014)), but we include the proof for completeness.

*Proof of Theorem 2.* The lower bound of $\Omega(RL/\sqrt{n})$ comes from standard lower bound for SCO without privacy constraints, hence we focus on the first term in the max. In the rest of the proof, we will be dealing with the class of linear loss functions and hence without loss of generality, we assume $R = L = 1$, and prove a lower bound of $\Omega\left(\sqrt{\frac{Ke^{\varepsilon}}{(e^{\varepsilon}-1)^2}} \cdot \frac{1}{\sqrt{n}}\right)$. Without loss of generality, we also assume $K$ is an even number. We will now use an algorithm $\mathcal{A}$ to construct an algorithm $\mathcal{A}'$ for discrete distribution estimation. The algorithm $\mathcal{A}'$ works as follows:

- We set the loss function to be independent of $X$, as $\ell(w; (X, Y)) = -\frac{1}{2}\left\langle w, e_Y - \frac{1}{K} \cdot \mathbf{1}_K\right\rangle$ where $e_Y$ is the one-hot vector with a one at location $Y$ and $\mathbf{1}_K$ is the $K$-dimensional all-ones vector. It is simple to verify that this loss function is 1-Lipschitz.
- Let $x_0 \in \mathcal{X}$ be a fixed element.
- Upon receiving $Y_1, \ldots, Y_n$ as inputs, $\mathcal{A}'$ runs $\mathcal{A}$ on $(x_0, Y_1), \ldots, (x_0, Y_n)$ to get an output $\hat{w}$.
- Output $\hat{\theta} = \frac{1}{K} \cdot \mathbf{1}_K + \alpha \cdot \hat{w}$, where $\alpha = \gamma\sqrt{K}$ and $\gamma = \Theta\left(\sqrt{\frac{e^{\varepsilon}}{(e^{\varepsilon}-1)^2}} \cdot \frac{1}{\sqrt{n}}\right)$ is from Theorem 3.

Since $\mathcal{A}'$ is simply a post-processing of $\mathcal{A}$, it satisfies (fully-interactive) $\varepsilon$-L-LDP. From Theorem 3, there exists $\theta \in \left\{\frac{1}{K} + \gamma, \frac{1}{K} - \gamma\right\}$ for some $\gamma = \Theta\left(\sqrt{\frac{e^{\varepsilon}}{(e^{\varepsilon}-1)^2}} \cdot \frac{1}{\sqrt{n}}\right)$ such that

$$\mathbb{E}_{\hat{\theta} \leftarrow \mathcal{A}'(\mathcal{D}_{\theta}^{\otimes n})}[\|\hat{\theta} - \theta\|_2] \geq \Omega\left(\sqrt{\frac{e^{\varepsilon}K}{(e^{\varepsilon}-1)^2}} \cdot \frac{1}{\sqrt{n}}\right). \tag{2}$$

We will use this to analyze the error of $\mathcal{A}$. To do so, let $P_{\theta}$ denote the distribution of $(x_0, Y)$ where $Y \sim \mathcal{D}_{\theta}$, and let $b = \frac{1}{\alpha} \cdot \left(\theta - \frac{1}{K} \cdot \mathbf{1}_K\right)$. Note that $\|b\|_2 = 1$. First, observe that

$$\mathcal{L}(w, P_{\theta}) = -\frac{1}{2}\mathbb{E}_{Y \sim \mathcal{D}_{\theta}}\left[\left\langle w, e_Y - \frac{1}{K}\mathbf{1}_K\right\rangle\right] = -\frac{1}{2}\left\langle w, \theta - \frac{1}{K}\mathbf{1}_K\right\rangle = -\frac{\alpha}{2}\langle w, b\rangle.$$

Thus, we have $w^* = b$. Furthermore, for all $w$ with $\|w\| \leq 1$,

$$\mathcal{L}(w, P_{\theta}) - \mathcal{L}(w^*, P_{\theta}) = \frac{\alpha}{2} - \frac{\alpha}{2}\langle w, b\rangle \geq \frac{\alpha}{4}\left(\|b\|_2^2 + \|w\|_2^2 - 2\langle w, b\rangle\right) = \frac{\alpha}{4}\|w - b\|_2^2.$$

Moreover, by setting $\hat{\theta} = \frac{1}{K} \cdot \mathbf{1}_K + \alpha \cdot \hat{w}$, we have $\|\hat{\theta} - \theta\|_2 = \alpha \|\hat{w} - b\|_2$. Combining these, we have

$$
\mathbb{E}_{\hat{w} \leftarrow \mathcal{A}(P_\theta^{\otimes n})}[\mathcal{L}(\hat{w}, P_\theta)] - \mathcal{L}(w^*, P_\theta) \geq \frac{1}{4\alpha} \mathbb{E}_{\hat{\theta} \leftarrow \mathcal{A}'(\mathcal{D}_\theta^{\otimes n})}[\|\hat{\theta} - \theta\|_2^2]
$$

$$
\text{(Jensen's Inequality)} \geq \frac{1}{4\alpha} \left( \mathbb{E}_{\hat{\theta} \leftarrow \mathcal{A}'(\mathcal{D}_\theta^{\otimes n})}[\|\hat{\theta} - \theta\|_2] \right)^2
$$

$$
\overset{(2)}{\geq} \Omega\left( \frac{1}{\sqrt{\frac{e^\varepsilon}{(e^\varepsilon - 1)^2} \cdot \frac{1}{\sqrt{n}} \cdot \sqrt{K}}} \right) \cdot \Omega\left( \sqrt{\frac{e^\varepsilon K}{(e^\varepsilon - 1)^2} \cdot \frac{1}{\sqrt{n}}} \right)^2
$$

$$
= \Omega\left( \sqrt{\frac{e^\varepsilon K}{(e^\varepsilon - 1)^2}} \cdot \frac{1}{\sqrt{n}} \right). \qquad \square
$$

## 5 Conclusion and Open Questions

In this work, we give a new algorithm and a lower bound for SCO with local Label DP. Our lower bound is tight for *all* regimes of $\varepsilon$, assuming that $n$ is sufficiently large. An obvious open question here is whether one can achieve tight bounds for *all $n$* as well. We remark that, even in discrete distribution estimation with $\ell_2$-error (which we reduce from), the known lower bounds (Ye & Barg, 2018; Acharya et al., 2023) are *not* tight. Specifically, for $\Theta\left( \frac{e^\varepsilon K^2}{(e^\varepsilon - 1)^2} \right) \geq n \geq \Theta\left( \frac{1}{(e^\varepsilon - 1)^2} \right)$, the lower bound from (Acharya et al., 2023) is $O\left( \sqrt[4]{\frac{e^\varepsilon}{(e^\varepsilon - 1)^2 n}} \right)$, which (to the best of our knowledge) is only tight up to polylogarithmic factors. In particular, there is a gap of $O(\sqrt[4]{\log K})$ between the best known upper bound from (Bassily, 2019) and the aforementioned lower bound. Furthermore, the algorithm from (Bassily, 2019) employs a projection technique which results in a bias, making it unsuitable for SGD algorithms in our SCO setting. (See also (Ghazi et al., 2024a), which analyzes a projection-based Label DP algorithm in the *central* model. Their analysis also yields a non-optimal dependency on $n$ due to the bias.)

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

## A   Variant of the Algorithm via $d$-Subset Selection Mechanism

In this section, we sketch an argument showing that we may replace the randomizer in Section 3 with the randomizer from (Ye & Barg, 2018; Wang et al., 2016) and still achieve the same asymptotic excess risk guarantee, albeit via slightly more involved calculations. Throughout this section, we assume $\varepsilon \leq \ln K$ for simplicity. (Otherwise, we may use the analysis for $\varepsilon = \ln K$ instead.)

Recall the *d-Subset Selection* (Ye & Barg, 2018; Wang et al., 2016), where $d \in [K-1]$ is a parameter. We use this to construct the $\varepsilon$-L-LDP *local randomizer* $\mathcal{R}_t$ with output space $\mathcal{S}_d = \binom{[K]}{d}$. For a private label $Y_t$, $\mathcal{R}_t(Y_t)$ outputs a subset $S_t \subseteq [K]$ of size $d$ where

$$\Pr[S_t = S] = \begin{cases} \frac{e^\varepsilon}{e^\varepsilon \cdot \binom{K-1}{d-1} + \binom{K-1}{d}} & \text{if } S \ni Y_t \\ \frac{1}{e^\varepsilon \cdot \binom{K-1}{d-1} + \binom{K-1}{d}} & \text{otherwise.} \end{cases} \qquad \forall S \in \binom{[K]}{d} \qquad (3)$$

Let $\gamma_d = \frac{1}{1+e^{-\varepsilon} \cdot \frac{K-d}{d}}$ and $\zeta_d = \frac{d-\gamma_d}{K-1}$. Notice that we have

$$\Pr[i \in S_t] = \begin{cases} \gamma_d & \text{if } i = Y_t, \\ \zeta_d & \text{otherwise.} \end{cases}$$

We can thus use a similar analyzer $\mathcal{A}_0$ as in Algorithm 1 except that the estimator is now

$$\hat{g}_t = \frac{1}{\gamma_d - \zeta_d} \left( \sum_{k \in [K]} \left( \mathbb{1}\{k \in S_t\} - \zeta_d \right) \cdot \nabla \ell(w_{t-1}, (X_t, k)) \right). \tag{4}$$

It is obvious to check that $\mathbb{E}[\hat{g}_t] = \nabla \ell(w_{t-1}, (X_t, Y_t))$. Thus, we are left to bound $\mathbb{E}\left[\|\hat{g}_t\|^2\right]$.

Throughout, we assume that $d \leq 2K/3$; this is the standard setting of parameters (and will be satisfied by our choice of $d$ below). Notice that

$$\frac{1}{\gamma_d - \zeta_d} = \frac{(K-1)(e^\varepsilon + \frac{K-d}{d})}{(K-d)(e^\varepsilon - 1)} \lesssim \frac{e^\varepsilon + K/d}{e^\varepsilon - 1}.$$

Furthermore, from Equation (4), we have

$$\|(\gamma_d - \zeta_d) \cdot \hat{g}_t\|^2 \leq 2\|(\mathbb{1}\{Y_t \in S_t\} - \zeta_d) \cdot \nabla\ell(w_{t-1}, (X_t, Y_t))\|^2$$

$$+ 2\left\|\left( \sum_{k \in [K] \setminus Y_t} \left( \mathbb{1}\{k \in S_t\} - \zeta_d \right) \cdot \nabla\ell(w_{t-1}, (X_t, k)) \right)\right\|^2. \tag{5}$$

For the first RHS term, observe that $\zeta_d \leq 1$ and thus, we simply have

$$\|(\mathbb{1}\{Y_t \in S_t\} - \zeta_d) \cdot \nabla\ell(w_{t-1}, (X_t, Y_t))\|^2 \leq L^2.$$

For the second RHS term, notice that, for all distinct $k, k' \in [K] \setminus Y_t$, we have

$$\mathbb{E}[\mathbb{1}\{k \in S_t\} - \zeta_d] = 0$$
$$\mathbb{E}[(\mathbb{1}\{k \in S_t\} - \zeta_d)^2] = \zeta_d(1 - \zeta_d)$$
$$\mathbb{E}[\langle \mathbb{1}\{k \in S_t\} - \zeta_d, \mathbb{1}\{k' \in S_t\} - \zeta_d\rangle] = \Pr[\{k, k'\} \subseteq S_t] - \zeta_d^2 =: \theta$$

Observe that $-\zeta_d^2 \leq \theta \leq 0$. We can bound the expectation of the second RHS term in (5) as follows:

$$\mathbb{E}\left[\left\|\left( \sum_{k \in [K] \setminus Y_t} \left( \mathbb{1}\{k \in S_t\} - \zeta_d \right) \cdot \nabla\ell(w_{t-1}, (X_t, k)) \right)\right\|^2\right]$$

$$= \left( \sum_{k \in [K] \setminus Y_t} \zeta_d(1 - \zeta_d) \cdot \|\nabla\ell(w_{t-1}, (X_t, k))\|^2 \right)$$

$$+ 2\left( \sum_{k \neq k' \in [K] \setminus Y_t} \theta \cdot \langle \nabla\ell(w_{t-1}, (X_t, k)), \nabla\ell(w_{t-1}, (X_t, k'))\rangle \right)$$

$$= \left( \sum_{k \in [K] \setminus Y_t} (\zeta_d(1 - \zeta_d) - \theta) \cdot \|\nabla\ell(w_{t-1}, (X_t, k))\|^2 \right) + \theta \cdot \left\|\sum_{k \in [K] \setminus Y_t} \nabla\ell(w_{t-1}, (X_t, k))\right\|^2$$

$$\leq (K-1)\zeta_d L^2$$

$$\leq dL^2,$$

where the first inequality is due to $-\zeta_d^2 \leq \theta \leq 0$, and the last inequality is due to the definition of $\zeta_d$. Combining the above inequalities, we get

$$\mathbb{E}[\|\hat{g}_t\|^2] \lesssim \left(\frac{e^\varepsilon + K/d}{e^\varepsilon - 1}\right)^2 \cdot dL^2.$$

Now, set $d = \lceil K/(2e^\varepsilon) \rceil$; note that $d = \Theta(1 + K/e^\varepsilon)$. Plugging this into the above, we get

$$\mathbb{E}[\|\hat{g}_t\|^2] \lesssim \left(\frac{e^\varepsilon}{e^\varepsilon - 1}\right)^2 \cdot \Theta(1 + K/e^\varepsilon)L^2 \lesssim \frac{e^\varepsilon(e^\varepsilon + K)}{(e^\varepsilon - 1)^2}L^2.$$

This is the same asymptotic bound as in Section 3 so this variant yields the same excess risk bound.

## B An Algorithm Using (Duchi et al., 2013)

We briefly discuss an $\varepsilon$-L-LDP algorithm based on a vector randomization algorithm of (Duchi et al., 2013), which is stated below. We note that the original paper (Duchi et al., 2013) only proves the following for $\mathcal{Q} = R^d$; however, it can be easily extend to any $d$-dimensional subspace $\mathcal{Q}$ by writing the input vector $v$ using the basis of $\mathcal{Q}$ and applying the original algorithm on the latter.

**Theorem 4** ((Duchi et al., 2013)). *Let $\mathcal{Q}$ be a (publicly known) $d$-dimensional subspace of $\mathbb{R}^m$ and $L > 0$. There is an $\varepsilon$-local DP randomizer such that, given a vector $v \in \mathcal{Q}$ with $\|v\|_1 \leq L$, outputs a vector $\hat{v}$ such that $\mathbb{E}[\|v - \hat{v}\|^2] \leq O\left(\frac{L^2 d}{\min\{1, \varepsilon^2\}}\right)$.*

The algorithm for SCO is exactly the same as Algorithm 1 except that, on Line 10, we obtain $\hat{g}_t$ via applying the above randomizer on $v = \nabla\ell(w_{t-1}, (X_t, Y_t))$ with $\mathcal{Q}$ being the span of $\{\nabla\ell(w_{t-1}, (X_t, k))\}_{k \in [K]}$. Note that $\mathcal{Q}$ here has dimension $d = K$. This gives the following result.

**Corollary 1.** *There exists a local $\varepsilon$-label DP algorithm $\hat{w}$ satisfying the following*

$$\mathbb{E}_{\hat{w} \leftarrow \mathcal{A}(P^{\otimes n})}[\mathcal{L}(\hat{w}, P)] - \mathcal{L}(w^*, P) = O\left(\frac{\sqrt{K}}{\min\{1, \varepsilon\}} \cdot \frac{RL}{\sqrt{n}}\right).$$

As mention earlier, this matches the excess risk in our main theorem (Theorem 1) in the high-privacy regime, but our bounds are better in the other two regimes. Furthermore, our algorithm is non-interactive whereas this algorithm is (sequentially) interactive.

