# OpenReview forum: "Convex Optimization with Local Label Differential Privacy: Tight Bounds in All Privacy Regimes"
_TMLR — Accepted by TMLR_

### Review · Reviewer_N2sj · 2026-02-19

**Summary Of Contributions:**

The paper studies differentially private stochastic convex optimization under local label privacy, where it is assumed the label is in [K]. The previous best known excess risk rate for this problem was $\frac{K}{\sqrt{n}\epsilon}$. The paper improves this to $\sqrt{\frac{K}{n\epsilon}} and shows this is tight. The main technical divergence from past work (in their upper bound) is the use of a different randomizer for privatizing the label. After this randomizer is established, it is plugged into SGD and analyzed in a fairly standard way. The lower bound follows from a reduction to a known lower bound.

**Audience:**

Yes

**Audience Explanation:**

If you accept the premise that label privacy is something we should be studying, I think this is a nice result and a perfect fit for TMLR. The problem studied in this work is fundamental enough that knowing the optimal rates is certainly beneficial.

My main concern with the paper is the very fact that it studies label privacy. The entire premise of machine learning is that the label can be predicted from the features. As such, an algorithm which only guards against revealing the labels but not the features does not actually provide a meaningful privacy guarantee for the labels. My understanding is that the community at large has also moved away from label privacy for these reasons.

I'm a bit torn, because outside of my issues with label privacy, I think this paper is an easy accept. That said, I am worried that publishing it lends credence to the idea that label private algorithms provide a meaningful privacy guarantee. Can the authors defend why label privacy is still worth studying?

**Broader Impact Concerns:**

I would request that the authors clarify in their work that the privacy protections offered by label differential privacy are not comparable to those traditionally associated with differential privacy. Specifically, label privacy does not protect the labels from being reconstructed by adversaries with enough side information, because anyone that has an accurate model and access to the feature can predict the label. I believe this statement is important to stop the ongoing misconception that protecting the label is simply a matter of applying differentially private mechanisms to that component of the data.

**Claims And Evidence:**

Yes

**Claims Explanation:**

Proofs are provided for all claims made. The technical content was fairly brief, so checking for correctness was straightforward.

**Requested Changes:**

If possible, the authors should provide a justification for scenarios where the privacy guarantees of label privacy are meaningful.

---

> ### Author Response · Authors · 2026-02-20
>
> We sincerely thank the reviewer for their review. We respectfully disagree that the privacy community has moved away from Label DP, or that it fails to provide a meaningful guarantee.
>
> To motivate Label DP, we discuss the following real scenarios:
>
> - **Computational Advertising:** The assumption that privacy-preserving algorithms must guard both features and labels holds for a centralized dataset where all elements are equally sensitive. In some cases, data is vertically partitioned, where the adversary (i.e., the model trainer) already owns the features; a nice example is in computational advertising (e.g., the [Google Chrome Privacy Sandbox Attribution Reporting API](https://privacysandbox.google.com/private-advertising/attribution-reporting)). The Ad network serving the impression already knows the user's context (i.e., the features). However, the subsequent purchase on an external site (i.e., the conversion label) is deemed sensitive. Label DP is a valid notion to protect this localized sensitive signal without injecting unnecessary noise into features that the Ad network already possesses.
>
> - **Medical Diagnostics:** In epidemiological/sociological learning applications, features such as age, gender, geographic location, are frequently semi-public or shared in open registries. However, the specific label (e.g., rare psychiatric diagnosis, stigmatized medical condition) is protected by HIPPA-like frameworks. Applying full DP (e.g., DP-SGD) to the entire high-dimensional patient features might obliterate the faint signals necessary for diagnosis. Label DP preserves the integrity of the public features while provably bounding the risk of an adversary confirming the specific, sensitive diagnosis.
>
> In addition to these, please note that the "labels" in our framework does not have to include only the labels, rather it can include any sensitive features (where $K$ would become the number of all possible combinations of these sensitive features).
>
> In our submission, for brevity reasons, we keep only a few references that are directly related to our results.  Please note that the literature on Label DP (or Semi-Sensitive DP) is vast and extant. Over the past two and a half years alone, there have been numerous recent papers at top conferences on the topic. Specifically, in addition to the two papers at COLT'24 and ICLR'24 cited in our submission, here are sample publications on Label DP at ICLR’25 [A], AISTATS’25 [B], S&P’25 [C] NeurIPS’24 [D], AdKDD’24 [E], ICLR’24 [F], NeurIPS’23 [G], ICLR’23 [H].
>
> We are happy to include the above discussion and these additional references in the revision.
>
> ## References
>
> A. Puning Zhao, Jiafei Wu, Zhe Liu, Li Shen, Zhikun Zhang, Rongfei Fan, Le Sun, Qingming Li:
> Enhancing Learning with Label Differential Privacy by Vector Approximation. ICLR 2025
>
> B. Yuheng Ma, Ke Jia, Hanfang Yang:
> Locally Private Estimation with Public Features. AISTATS 2025: 55-63
>
> C. Saeed Mahloujifar, Chuan Guo, G. Edward Suh, Kamalika Chaudhuri: Machine Learning with Privacy for Protected Attributes. SP 2025: 2640-2657
>
> D. Róbert Busa-Fekete, Travis Dick, Claudio Gentile, Andrés Muñoz Medina, Adam Smith, Marika Swanberg: Auditing Privacy Mechanisms via Label Inference Attacks. NeurIPS 2024
>
> E. Lynn Chua, Qiliang Cui, Badih Ghazi, Charlie Harrison, Pritish Kamath, Walid Krichene, Ravi Kumar, Pasin Manurangsi, Nicolas Mayoraz, Hema Venkata Krishna Giri Narra, Steffen Rendle, Amer Sinha, Avinash V. Varadarajan, Chiyuan Zhang: Training Differentially Private Ad Prediction Models With Semi-Sensitive Features. AdKDD@KDD 2024
>
> F. Badih Ghazi, Yangsibo Huang, Pritish Kamath, Ravi Kumar, Pasin Manurangsi, Chiyuan Zhang: LabelDP-Pro: Learning with Label Differential Privacy via Projections. ICLR 2024
>
> G. Ashwinkumar Badanidiyuru Varadaraja, Badih Ghazi, Pritish Kamath, Ravi Kumar, Ethan Leeman, Pasin Manurangsi, Avinash V. Varadarajan, Chiyuan Zhang:
> Optimal Unbiased Randomizers for Regression with Label Differential Privacy. NeurIPS 2023
>
> H. Badih Ghazi, Pritish Kamath, Ravi Kumar, Ethan Leeman, Pasin Manurangsi, Avinash V. Varadarajan, Chiyuan Zhang: Regression with Label Differential Privacy. ICLR 2023
>
> I. Róbert Busa-Fekete, Travis Dick, Claudio Gentile, Andrés Muñoz Medina, Adam Smith, Marika Swanberg: Auditing Privacy Mechanisms via Label Inference Attacks. NeurIPS 2024

---

> > ### Comment · Reviewer_N2sj · 2026-02-20
> >
> > Thank you for your very well written rebuttal. However, I believe it does not address the fundamental issue that it is impossible to guard the secrecy of the label without also protecting the secrecy of the feature. Unless the label and feature are uncorrelated (in which case we're probably not doing machine learning) then knowing the feature gives someone a way to predict the label with non-negligable confidence. In your ad example, perhaps the model trainer does not currently have the ability to predict the label, but with enough side information (which does not have to be a noisy version of that specific label) they could in the future predict the label.
> >
> > That said, you clearly demonstrate in your rebuttal that there is indeed ongoing work on label privacy, for better or worse. I will  recommend acceptance **conditional** on the paper adding a disclaimer that the privacy protections provided by label privacy are not comparable to those generally associated with DP (i.e. protection under arbitrary amounts of side information). I have also updated my review to reflect this.

---

> > > ### Author Response · Authors · 2026-02-22
> > >
> > > Thank you for your follow-up response and for your quick and continued engagement with our work. We appreciate your recommendation for acceptance.
> > >
> > > However, we must clarify that Label DP *does* protect against arbitrary amounts of side information. In Label DP, the public features themselves can be viewed as the adversary's side information. Because Label DP shares the same mathematical foundation as standard DP, it guarantees that *even when conditioned on arbitrary side information (including the features)*, the adversary's posterior belief about the specific label instance does not meaningfully change as a result of the algorithm's output.
> > >
> > > More generally, we want to clarify a fundamental distinction between *population-level statistical inference* and *individual-level privacy compromise*.
> > >
> > > It is true that if features ($X$) and labels ($Y$) are highly correlated, an adversary can predict $Y$ from $X$ with high confidence; as you point out, learning this correlation is the exact purpose of machine learning and statistical analysis. However, learning generalized statistical facts about a population does not constitute a privacy breach. A privacy framework is designed to prevent observers from definitively learning the *exact, ground-truth data* of a *specific* individual used in the training set. We note that this distinction has been discussed many times in the privacy literature including in the blog posts [[A](https://differentialprivacy.org/inference-is-not-a-privacy-violation/),[B](https://github.com/frankmcsherry/blog/blob/master/posts/2016-06-14.md)], by some of the inventors of DP.
> > >
> > > Here is a simple example to demonstrate our point. Consider an election where a candidate wins a landslide victory in a specific county, capturing 99% of the votes. If we know Alice lives in that county (her public feature), we can predict her vote (the sensitive label) with 99% accuracy. This is a highly accurate population-level statistical inference. However, learning her *actual, individual ballot* would be a catastrophic privacy failure.
> > >
> > > Label DP provides Alice with mathematical plausible deniability. Even though the adversary strongly suspects her vote based on her public features, the Label DP mechanism ensures they can never be 100% certain of the *actual vote* she cast in the election. The privacy guarantee prevents the deterministic memorization of her specific, realized outcome, regardless of how predictable the population-level trend makes it.
> > >
> > > Because the distinction between statistical inference and privacy protection is a foundational premise of DP—rather than a limitation unique to Label DP—we respectfully believe that adding a disclaimer stating Label DP lacks side-information guarantees would be technically inaccurate.
> > >
> > > We thank you again for your time, your thorough review, and for helping us refine the positioning of our work.

---

> > > > ### Comment · Reviewer_N2sj · 2026-03-02
> > > >
> > > > Thank you for your ongoing discussion. I agree that the statement you made about label DP and side information is correct in some sense, but it is not correct for the scenario you are studying. We are implicitly assuming that the training algorithm uses the feature vector associated with label y. This process of the selecting the correct feature vector is not a differentially private mechanism of the label. Indeed, if the data domain is such that a there is a feature vector with only 1 possible label, recovering the feature vector may allow an adversary to recover the label w.p. 1 (i.e. no plausible deniability). To put this another another way, yes label DP protects against an adversary with enough side information to learn the true predictor f(x)=y. But it stops protecting the label, $y$, if the adversary learns that $x$ was in the dataset. Since we're not assuming DP w.r.t. to the features, we cannot assume the adversary does not know that $x$ was in the dataset.
> > > > I suppose you could assume the feature vectors are selected in some differentially private manner, but clearly that is not the intention behind the scenario you study in the paper.
> > > >
> > > > I do think the voter example is still compelling, and as stated previously, I buy your point that label DP does have some practical relevance. I still maintain that there is misconception about what label DP means from a privacy perspective, which should be clarified in papers that study it.

---

> > > > > ### Author Response · Authors · 2026-03-02
> > > > >
> > > > > Thank you the reviewer for the response.
> > > > >
> > > > > We are still not totally sure if we totally understand your concern but we can add the following as a footnote in the paragraph before Definition 1: "It should be stressed that Label DP does \emph{not} provide any privacy protection against the features $X_i$. Thus, it is only appropriate to apply e.g. when the features are considered publicly available." Would that be sufficient to address your concern?
> > > > >
> > > > > If not, then can you please help add more detail on a problematic scenario where all the features are publicly known (to the adversary)? In this case, we can try to understand the issue more deeply and add a more appropriate remark in the revision. Thank you.

---

> > > > > > ### Comment · Reviewer_N2sj · 2026-03-02
> > > > > >
> > > > > > Perhaps I can lay it out in the following example. Lets say we have an agent, Alice, who has access to the data, $D=(Y_1,X_1),...,(Y_n,X_n)$. Alice gives the feature vectors to a model trainer, Bob. Alice also gives Bob privatized versions of the labels. Bob would like to release a trained model, $w$, and claim it preserves the privacy of the labels.
> > > > > >
> > > > > > When does the statistic, $w$, protect the privacy of the labels? Differential privacy requires the distribution of $w$ is indistinguishable from the distribution I would get from changing a label. I'm saying that applying DP mechanisms to the labels does not imply this indistinguishability. This is because if I were to change a label, I also change the feature vector that Alice gave Bob, because Alice has agreed to always give Bob the correct feature vector. The process of giving Bob the correct feature vector is not a differentially private statistic of the label. Thus, the model $w$ also may not be a differentially private statistic of the label.
> > > > > >
> > > > > > Certainly once you treat the information that $X_1,...,X_n$ are in D as public information, then label private mechanisms do not leak much additional information about whether $Y_1,...,Y_n$ are in D. But this is a non-sensical idea, because if I know $X_1,...,X_n$ are in D, someone with a good predictor could already know that $Y_1,...,Y_n$ are in $D$ with probability $1$ (if we assume realizability).
> > > > > >
> > > > > > To emphasize further, the problem here is that not that an adversary might learn what the feature vector, $X$, associated with a label $Y$, is (for example). The problem is that the adversary might learn that feature vector was in the dataset. Since label privacy does not prevent an adversary from learning $X$ was in the dataset, if the adversary was able to learn the true predictor, the adversary now knows $Y$ is in the dataset.

---

> > > > > > > ### Author Response · Authors · 2026-03-02
> > > > > > >
> > > > > > > We thank the reviewer for the explanation. Can the reviewer please also confirm whether they are happy with the footnote we proposed to add? (If not, then please feel free to suggest any specifics.)
> > > > > > >
> > > > > > > We believe we understand the reviewer's concern, and that this concern does not occur in the case where the $X_i$'s are public (because the adversary does not learn any extra information from the algorithm in this example). In particular, to go back to our voting example, if the candidate wins with 100% (instead of 99%) of the vote. Then, any adversary that knows that Alice lives in that county (her public feature) will be able to deduce that Alice votes for this candidate--regardless of whether we run any (DP or not DP) algorithm on it.
> > > > > > >
> > > > > > > Thank you again.

---

> > > > > > > > ### Comment · Reviewer_N2sj · 2026-03-03
> > > > > > > >
> > > > > > > > So long as the comment raises some sort of awareness of the failure case of label DP I will not nitpick. If I was writing it this is what I would write.
> > > > > > > >
> > > > > > > > "It should be stressed that Label DP does \emph{not} provide any privacy protection for the features $X_i$. As such, label DP only limits an adversaries ability to infer the label beyond what is possible for an adversary with access to the feature vectors. Users of label DP should be aware that in some cases, leaking the fact that the feature vectors were in the dataset may already give an adversary too much power to predict the labels in the dataset."

---

> > > > > > > > > ### Author Response · Authors · 2026-03-03
> > > > > > > > >
> > > > > > > > > We've added a variant of your suggested paragraph as footnote 1 in the revised version. Thank you again.

---

### Review · Reviewer_1jja · 2026-02-19

**Summary Of Contributions:**

The article makes improvements on the problem of stochastic convex optimization under local label differential privacy. Namely, the authors use a better sanitation mechanism compared to the randomized response one that was previously proposed, which allows the authors to obtain the correct scaling in the number of classes. The authors also propose a matching lower bound.

Strengths:
- The paper is well written and its message is clear.
- The contribution is clearly identifiable.
- The authors identified how to combine the different (rather standard) techniques used in the article to obtain a better estimation rate.
- The connection of the proposed mechanism to other local-DP mechanisms that are made in the appendix is interesting.

Weaknesses:
- I do not see many negative points to the article as, being on the theory side and proving new bounds for the problem, it fulfills its job. However, I would have liked a small numerical comparison to the previous SOTA in order to know if the claimed improvement is of interest only for the theoreticians or if it also has implications for the practitioner.

**Audience:**

Yes

**Audience Explanation:**

The problem that is studied in this article is of interest to the privacy-preserving machine learning community. The authors present what is, to the best of my knowledge, the first minimax-optimal private mechanism to solve the problem.

**Claims And Evidence:**

Yes

**Claims Explanation:**

Overall, the paper, despite being on the short side, is well written and goes to the point. It presents a clear improvement on a problem of interest to the community of privacy-preserving machine learning. The claimed results appear to be properly proved.

**Requested Changes:**

I am happy with the current state of the article. I think that a small numerical section would improve the article, but this is only a recommendation.

---

> ### Author Response · Authors · 2026-02-20
>
> Thank you for your review and recommendation. To maintain the analytical focus of the current study, we believe a numerical analysis would be better suited for future research. We will include a statement to this effect in the revised manuscript.

---

### Review · Reviewer_zhHa · 2026-03-03

**Summary Of Contributions:**

This paper considers a convex optimization problem under local label differential privacy (L-LDP). This paper improves the previous upper bounds for this problem (by Ghazi et al. 2021) through their Algorithm 1. The idea behind this algorithm is neat and compact, and based on the list of prior work on local DP for hystogram estimation referred at the beginning of Section 3. The proof of the upper-bound theorem is well written and (seemingly) correct.

The Authors also provide a matching lower bound for a wide range of privacy parameters $\varepsilon$ and number of samples $n$ (with respect to the number of label classes $K$). This involves exhibiting a data distribution for which no $\varepsilon$ L-LDP algorithmcan have an error below the given threshold. This is done adapting the result on discrete distribution estimation by Acharya et al 2023).

The setting (in particular Lipschitz loss and bounded parameter space) is standard in the literature.

**Audience:**

Yes

**Audience Explanation:**

While being a quite specific setting (private convex optimization with local label DP) I believe the community of private optimization to find at least some sub-parts of this work interesting, from the application of private hystograms in optimization to the lower bound results based on Acharaya et al. 2023.

**Claims And Evidence:**

Yes

**Claims Explanation:**

Yes. The evidence consists in the proofs of the main theoretical results (Theorem 1 and 2). The proofs are compact and included in the main body of the paper.

**Requested Changes:**

I recommend acceptance without changes. The Authors could consider changing the $n \geq \Theta (\cdot)$ in the statement of Theorem 2 with a $\Omega(\cdot)$, and also add "the true worst-case cost of L-LDP ..." in the end of Section 1.

---

> ### Author Response · Authors · 2026-03-03
>
> We thank the reviewer for their review. We've added "worst-case" to the last sentence in Section 1 as the reviewer suggested. However, we decided to keep $\Theta$ (instead of $\Omega$) as readers sometimes confuse the asymptotic notations in quantified variables.

---

### Decision · Action_Editor_6rzR · 2026-03-26

**Recommendation:** Accept as is

**Audience:**

Yes

**Audience Explanation:**

The paper resolves a theoretical gap by establishing the minimax optimal rate for private convex optimization under label DP.  This will be of interest to researchers in DP and theoretical machine learning.

**Claims And Evidence:**

Yes

**Claims Explanation:**

The paper addresses a fundamental problem in differential privacy (DP): stochastic convex optimization under local label DP. The paper gives a non-interactive algorithm that improves the dependency on the label space size $K$ from linear $O(K)$ to square root $O(\sqrt{K})$. A matching information-theoretic lower bound is given, showing that the achieved rate is minimax optimal.

All three reviewers agreed that the proofs are correct, the paper is well-written, and the message is clear. All three reviewers recommend Accept.

The review process included a constructive debate regarding the value of label DP in general. This valuable discussion was lead by reviewer N2sj who argued that label DP is of "limited interest" because if features are public and highly correlated with labels, an adversary can simply predict the label from the features. The authors countered this by highlighting real-world applications where features are already known but labels are sensitive and where label DP seems to be a good fit. To address reviewer N2sj's concern about potentially misleading readers, the authors added a footnote on page 1 which states that Label DP does not protect features and that if features perfectly determine a label, no real privacy is afforded. All in all, I believe this public exchange can serve as a useful reference for the community when trying to figure out the nature of the protection label DP provides.